# Adaptation of the Texas Christian University Organisational Readiness for Change Short Form (TCU-ORC-SF) for use in primary health facilities in South Africa

Carrie Brooke-Sumner  ,[1,2] Petal Petersen-Williams,[1,3] Emma Wagener,[4] Katherine Sorsdahl,[2] Gregory A Aarons,[5,6] Bronwyn Myers[1,3]

For numbered affiliations see end of article.

**Correspondence to**
Dr Carrie Brooke-Sumner; carrie.brooke-sumner@mrc.ac.za

## ABSTRACT

**Objectives** The Texas Christian University Organisational Readiness for Change Scale (TCU-ORC) assesses factors influencing adoption of evidence-based practices. It has not been validated in low-income and middle-income countries (LMIC). This study assessed its psychometric properties in a South African setting with the aim of adapting it into a shorter measure.

**Methods** This study was conducted in 24 South African primary healthcare clinics in the Western Cape Province. The TCU-ORC and two other measures, the Organisational Readiness to Change Assessment (ORCA) and the Checklist for Assessing Readiness for Implementation (CARI) were administered. The questionnaire was readministered after 2 weeks to obtain data on test–retest reliability. Three hundred and ninety-five surveys were completed: 281 participants completed the first survey, and 118 recompleted the assessments.

**Results** We used exploratory factor analysis (EFA) to identify latent dimensions represented in the data. Cronbach's alpha for each subscale was assessed and we examined the extent to which the subscales and total scale scores for the first and retest surveys correlated. Convergent validity was assessed by the correlation coefficient between the TCU-ORC, ORCA and CARI total scale scores. EFA resulted in a three-factor solution. The three subscales proposed are Clinic Organisational Climate (8 items), Motivational Readiness for Change (13 items) and Individual Change Efficacy (5 items) (26 items total). Cronbach's alpha for each subscale was >0.80. The overall shortened scale had a test–retest correlation of r=0.80, p<0.01, acceptable convergent validity with the ORCA scale (r=0.56, p<0.05), moderate convergence with the CARI (r=39, p<0.05) and strong correlation with the original scale (r=0.79, p<0.05).

**Conclusions** This study presents the first psychometric data on the TCU-ORC from an LMIC. The proposed shortened tool may be more feasible for use in LMICs.

**Trial registration number** Results stage. Project MIND trial. Pan-African Clinical Trials Registry. PACTR201610001825405.

### Strengths and limitations of this study

► This study presents a pragmatic tool for assessing organisational readiness for change in primary health facilities in South Africa.

► Factor structure and psychometric properties including temporal stability and convergent validity are presented indicating usefulness of this adapted tool in assessing readiness for change and enabling innovation adoption.

► Adaptation of this tool is a based on data collected from staff in primary health facilities in South Africa indicating its potential for other nurse-led and managed primary health settings.

► Data collected reflect the sociodemographic and organisational context of health facilities and contextual adaptation may be required for other settings.

## BACKGROUND

The treatment gap for mental disorders in South Africa, like other low-ncome and middle-income countries (LMICs) and high-income countries (HICs), is well documented, with increasing attention being directed towards addressing this gap by providing mental health services in primary care. With the reality of constrained budgets for health and human resource limitations in South Africa and globally, task-sharing of interventions to address mental disorders is gaining traction, whereby specific mental health services are delegated to non-specialist cadres with appropriate support and onward referral.[1 2] This approach is supported globally[3 4] as a strategy for building health system resilience[5 6] and is being investigated in a number of effectiveness and implementation studies in primary healthcare (PHC) settings in South Africa.[2 7–9] Little is known about the readiness and capability of many

BMJ

primary health facilities to implement this innovation, and there is great variation in implementation factors across facilities. Previous work by our group has identified patterns of factors associated with capability for implementing mental health counselling at primary care facilities in the Western Cape province. These included good patient information systems, organised services and a strong management team (all supporting the provision of patient-centred cared).[10] The variability across facilities observed in this work indicates the need for feasible ways to assess readiness and identify barriers to implementation for these service innovations.

As the introduction of task-shared interventions can be complex, some interventions fail to be implemented despite a supportive policy environment.[11] There may be several reasons for implementation failure in this context.[12] For example, psychosocial interventions can be complex in nature (eg, multiple sessions, psychological content); community contexts where patients live may challenge participation in sessions (eg, poverty, violence)[13 14]; and importantly, the organisational setting of primary care facilities presents a complex environment for introducing mental health and other behavioural health innovations (eg, heavy workloads, resource shortages, skills limitations, poor staff morale).[10 15–21] Innovations need to be implemented by individuals embedded in complex organisations and difficult social contexts.[22–24] This may help explain why many effective interventions do not transition to sustained routine practice in primary health organisations,[25 26] and why reducing the evidence-to-practice gap remains a priority for health services research.

Organisational readiness for change (ORC) provides a theoretical framework for conceptualising factors supporting or hindering initial adoption of new evidence-based practices. ORC, as defined by Weiner, is a shared psychological state that relies on organisation members' motivation to change (change commitment) and belief in their own capacity to change (change efficacy).[27] Commonly identified components contributing to this shared state include organisational dynamics, climate and culture (particularly leadership from management), and individual members' characteristics and readiness to accept change, including attitudes to evidence based practices.[27–31] Assessing ORC enables implementers to identify and develop approaches to reduce factors hindering innovation adoption and strengthen those factors that support adoption. In South Africa, constraints to innovation adoption include lack of resources, high workloads, inadequate staff skills, high levels of staff turnover and complex interpersonal and hierarchical relationships.[14 19 32 33] The assessment of ORC in South African settings has substantial potential to enable identification and mitigation of these constraints.

A variety of instruments for assessing ORC have been developed[34] and used in HICs. Assessment of ORC in LMICs is limited to date but may have important potential for improving implementation of quality healthcare.

The resources, operations and outer system and inner organisational contexts and processes are likely to differ between LMICs and HICs. This study aimed to address this research gap through contextually adapting an instrument for assessing ORC in primary health settings in South Africa.[35] The Texas Christian University ORC (TCU-ORC) scale[36] is among the most widely used of the ORC scales and has been tailored for use in health services research and with adaptation may be suitable for use in LMICs. Although there is good evidence for the instrument's validity in other settings,[34] and it has been used in South Africa,[37] little is known about the psychometric properties in LMICs and the South African context in particular. The TCU-ORC is a comprehensive measure of ORC assessing both individual psychological and organisational and health system influences on ORC. However, with 125 items, its length may make it unpractical for routine use in busy and under-resourced services. There are continuing calls for development and testing of brief and pragmatic measures (eg, low burden, important to stakeholders, actionable, psychometrically strong, related to theory) and such measures are more likely to be used in research and practice.[38 39]

This study was nested within a study known as Project MIND, a cluster randomised controlled trial evaluating two approaches to resourcing the integration of task-shared counselling for patients with depression and/or hazardous or harmful alcohol use.[2] The Project MIND counselling programme comprises three-sessions of motivational interviewing and problem-solving therapy delivered by trained lay counsellors.[2 40] The integration of this service constitutes a change to the basket of health services currently available within primary care clinics, providing an opportunity for contextually validating the TCU-ORC. This study aimed to assess the psychometric properties of the TCU-ORC in South African primary care clinics and reduce the number of items to a smaller, more manageable and relevant tool for this setting.

## METHODS

Methods for the Project MIND trial[2] and this adaptation substudy[35] are fully described elsewhere. Briefly, this study involved administering a set of ORC assessments to providers within Project MIND trial sites at two time points with the purpose of establishing the internal consistency, temporal stability and construct validity of the long-form TCU-ORC questionnaire.

### Participants and procedures

Data collection took place from July to December 2018 at the 24 facilities that participated in the MIND trial. Following a workshop that introduced the study in the facilities, health service employees (managers and staff) were requested to complete a paper based, English language self-report survey, after granting written informed consent. English is the official language of business of the South African health system.

Participants at workshops included facility managers, operational managers, medical officers, psychiatric nurses, general nursing staff from the chronic disease care team and lay counsellors. We aimed to secure participation of 12 staff members from each facility (288 participants in total), however, the number of participants per workshop was at the discretion of the facility manager based on staff availability. Between 1 and 2 weeks after the workshop, Project MIND fieldworkers asked a subsample of participants (convenience sample) from each facility to recomplete the paper-based survey comprising the ORC measures to enable assessment of test–retest reliability. These fieldworkers also approached nursing staff directly to fill in the questionnaire to gain enough responses for the first survey. In total, 395 surveys were completed: 281 participants completed the first survey and 118 participants completed both the first and test–retest survey.

## Measures

The survey comprised three measures of ORC; the TCU-ORC,[36] the Organisational Readiness to Change Assessment (ORCA)[41] and the Checklist for Assessing Readiness for Implementation (CARI).[42]

The four domains of the TCU-ORC (125 items total) measure Motivational Readiness for Change (Subscale A: covering staff's readiness, efficacy and support needs for implementing new practices; 33 items; eg, clinical staff at your programme needs guidance in matching client needs with services); Institutional Resources (Subscale B: adequacy of infrastructure, training and management; 31 items; eg, much time and attention are given to staff supervision when needed); Staff Attributes (Subscale C: staff skills, management planning and leadership; 31 items; eg, learning and using new procedures are easy for you) and Organisational Climate (Subscale D: shared goals, communication structure, work environment; 30 items; eg, the heavy staff workload reduces the effectiveness of your programme). Items are scored on a five-point Likert scale ranging from strongly disagree to strongly agree. In development, these subscales showed acceptable reliability (alpha >0.70 subscale A; alpha >0.60 subscale B; alpha >0.50 subscale C; alpha >0.50 subscale D).[36]

The Organisational Readiness For Change Assessment (ORCA)[41] comprises 50 items across three scales. These assess[1] strength of evidence for the change to be introduced (nine items; eg, the proposed change to the service delivery of the facility is supported by clinical experience with patients in other healthcare systems)[2]; quality of the organisational context (22 items; eg, in general in my facility, when there is agreement that change needs to happen we have the necessary support in terms of budget or financial resources) and[3] capacity for organisational facilitation of the change (19 items; eg, communication will be maintained through regular meetings covering the intervention). Items are scored on a five-point Likert scale ranging from strongly disagree to strongly agree. There is evidence for the overall reliability and factor structure of the ORCA (alpha 0.74, 0.85 and 0.95 for the subscales 1, 2 and 3, respectively).[41]

The CARI checklist[42] includes 36 items that assess: (1) organisational capacity (five items; eg, to what extent do you think technical assistance is available for the intervention being implemented); (2) organisational capacity (four items; eg, to what extent do you think financial resources are adequate and available to introduce and sustain the intervention); (3) functional considerations (four items, eg, to what extent do you think there is a system in place to share patient outcomes with staff and management), (4) organisational culture/climate (five items, eg, to what extent do you think there are open lines of communication in place in this facility; (5) senior leadership (five items; eg, to what extent do you think senior leadership is convinced of the value of this intervention; (6) staff capacity (five items; eg, to what extent do you think staff show they are ready for this type of practice change); (7) implementation plan (three items; eg, to what extent do you think there is intent to monitor fidelity of the intervention and (8) training; (five items, eg, to what extent do you think adequate time will be set aside for training, reflection and practice). This instrument was selected because it contains items that are broadly similar in content and wording to the TCU-ORC items although data on its psychometric properties remain to be published. Neither of the additional instruments have been validated in LMIC to our knowledge.

Twenty-three items across the four subscales of the TCU-ORC were removed prior to the assessment as they were not appropriate for the South African PHC context, (see online supplemental table 1) for item removal based on discussion with the research group), leaving 104 items in the ORC for further assessment. Some of the removed items related to computer availability. While staff in primary care settings are meant to have access to computers and the internet, in practice this is often not the case with staff having limited access to computers where these are available. Minor editing of items was carried out to improve readability and appropriateness for clinic staff (eg, 'your programme' was edited to read 'your facility').

## Analyses

Stata V.16 statistical software was used for analysis. Cases with >50% missing data were removed.[43] Missing data remaining in the data set were excluded from subsequent analysis using listwise exclusion. Normality tests were conducted for all measures using the Shapiro-Wilks test in combination with visual inspection of the normality plots. We used exploratory factor analysis, specifically principal axis factoring (PAF) to identify latent dimensions represented in the variables.[44 45] The factor solution was rotated using oblique oblimin rotation, which is suited when factors are possibly related, but measure different aspects of a construct.[46] Factor extraction was determined on the basis of the theoretical structure of the TCU-ORC,[36] as well as the Kaiser K0 rule for common

factors, in combination with Horn's parallel analysis.[47–49] Decisions around item retention were guided by item communalities and loading of items onto factors. Items with communalities of at least 0.70 are considered highly desirable.[50] Although items with loadings greater than 0.40 are often retained,[51] the decision was made to include items with factor loadings above 0.60 for parsimony due to the large number of items and to allow for a more stable factor solution.

Next, internal consistency of each subscale was assessed by calculating Cronbach's alpha of the reduced item subscales. Internal consistency is considered acceptable at values above 0.70 for generalisability.[52] To assess temporal stability, we examined the extent to which the subscales and total scales for the first and retest surveys correlated with each other, with a correlation coefficient above 0.70 considered adequate. Convergent validity was assessed by calculating the correlation coefficient between the TCU-ORC, ORCA and CARI measures for the total scales.[36 41 42]

## RESULTS

After removal of cases with less than 50% data, a total 275 participants with adequate responses who had completed the first survey and 118 participants who had completed both the first and retest survey were included in the analyses. Participants' sociodemographic characteristics are summarised in table 1. The sample comprised predominantly female nursing staff, reflecting the staffing profile of primary clinics in South Africa. A large proportion had worked at their current facility for more than 5 years and the majority had been in their profession for more than 5 years (table 1).

### Exploratory factor analysis
#### Assumption testing for factor analysis
Normality of the TCU-ORC was established by a non-significant Shapiro-Wilk result (p=0.11) and visual inspection of the Histogram and the Q-Q plot. Bartlett's test of sphericity indicated that correlations between items were sufficiently large for factor analysis ($\chi^2$(5356)=12 099.13, p<0.01). The Kaiser-Meyer-Olkin (KMO) measure of sampling adequacy was found to be acceptable with a value of 0.70.[53]

#### Factor extraction
The initial extraction was run with the Kaiser K0 rule specified, retaining factors with eigenvalues greater than 0. This identified 74 factors. A parallel analysis (PA) was run in Stata, using Glorfeld's (1995) extension and using the 95th and 99th percentile to compare eigenvalues in order to increase sensitivity.[49] The 95th percentile suggested retaining 14 factors, and the 99th percentile analysis resulted in 15 factors.

When conducting a PA, factors with a large number of variables are prone to inaccuracies due to poorly defined factors.[54] As this measure contains a large number of variables, it is likely that the results of the PA over extracted

**Table 1** Sociodemographic data from PHC facility-based staff

| Item | Category | Frequency | Percentage |
|------|----------|-----------|------------|
| Gender | Female | 214 | 88.4 |
| | Male | 28 | 11.6 |
| Age | 20–30 | 14 | 5.3 |
| | 31–40 | 72 | 27.2 |
| | 41–50 | 73 | 27.5 |
| | 51–60 | 84 | 31.7 |
| | 61–65 | 22 | 8.3 |
| Ethnicity | Black African | 136 | 50.2 |
| | Coloured | 117 | 43.2 |
| | White | 13 | 4.8 |
| | Indian/Asian | 3 | 1.1 |
| | Other | 2 | 0.7 |
| Time at facility | 0–6 months | 22 | 8.1 |
| | 6–11 months | 20 | 7.4 |
| | 1–3 years | 61 | 22.5 |
| | 3–5 years | 48 | 17.7 |
| | More than 5 years | 120 | 44.3 |
| Education level | Completed high school | 62 | 23.0 |
| | Some tertiary education | 65 | 24.1 |
| | Bachelor's degree | 51 | 18.9 |
| | Master's degree | 7 | 2.6 |
| | Doctoral degree | 3 | 1.1 |
| | Diploma in healthcare | 74 | 27.3 |
| | Other | 8 | 3.0 |
| Profession* | Medicine | 25 | 9.1 |
| | Social work | 6 | 2.1 |
| | Counselling | 62 | 22.7 |
| | Psychology | 2 | 0.7 |
| | Nursing | 149 | 54.8 |
| | Administration | 30 | 11.0 |
| | Other | 18 | 6.6 |
| Time in profession | 0–6 months | 7 | 2.6 |
| | 6–11 months | 7 | 2.6 |
| | 1–3 years | 36 | 13.4 |
| | 3–5 years | 25 | 9.4 |
| | More than 5 years | 192 | 72.0 |

*Some participants selected working in more than one profession.
PHC, primary healthcare.

factors. The decision was made to specify four factors, according to these results and the theoretical basis of the original TCU-ORC.[36] After the initial extraction of four factors, we observed that multiple items from two subscales loaded onto one factor. Visual inspection of the scree plot confirmed three distinct points above the point of inflection,[55] suggesting a three-factor solution. Subsequent analyses were based on the specification of a three-factor solution.

Factor loadings of the final factor structure are presented in table 2. Eleven extractions were performed overall to reach the final three factor solution. Items were removed one at a time in sequential factor analysis runs.[51] The final factor solution explained 72.4% of shared variance. We removed items with cross-loadings of 0.3 or above.[51] Sampling adequacy using the KMO Measure was assessed for each item as a postestimation after each extraction. In each extraction, items with loadings less than 0.60 were removed. The decision to remove items was also made with the consideration and assessment of how removing items affected the Cronbach's alpha for each subscale. The final three factor solution consists of 26 items, having removed 99 items and therefore providing a simplified, shortened version of the scale (table 3).

Factor 1: 'Clinic Organisational Climate' consists of 13 items, with loadings between 0.61 and 0.72. This factor is made up of items from the original Institutional Resources, Staff Attributes and Organisational Climate subscales. The revised subscale had a Cronbach's alpha of .90 and correlated with the original 'Organisational Climate' subscale ($r$=0.82, p<0.05).

Factor 2: 'Motivational Readiness for Change' consists of eight items, with loadings between 0.64 and 0.78. The items in this subscale were all originally in the TCU-ORC's Motivational Readiness for Change subscale. This subscale had good reliability with a Cronbach's alpha of 0.93 and correlated with the original 'Motivational Readiness for Change' subscale ($r$=0.79, p<0.05).

Factor 3: 'Individual Change Efficacy' consists of five items, with factor loadings between 0.62 and 0.76. The items in this subscale all originated from the 'Staff Attributes' subscale. This subscale had a Cronbach's alpha of .83 and correlated with the original 'Individual Change Efficacy' subscale ($r$=0.57, p<0.05).

Overall, the 26-item version of the TCU-ORC was strongly correlated with the original scale ($r$=0.79, p<0.05).

## Temporal stability
A test–retest correlation coefficient between the initial and follow-up versions of the TCU-ORC was conducted, using only those items retained after PAF for both administration versions. Temporal stability was assessed for the overall scale as well as for each subscale. The correlation results are presented in table 4. Subscales show adequate temporal stability for Clinic Organisational Climate ($r$=0.87, p<0.01), Motivational Readiness for Change ($r$=0.60, p<0.01), with exception of the Individual Change Efficacy subscale ($r$=0.53, p<0.01), Overall, the shortened scale was found to have good consistency across time with a test–retest correlation of $r$=0.80, p<0.01.

## Convergent validity
The reliability coefficients of criterion measures were good for the ORCA ($\alpha$=0.96) and CARI ($\alpha$=0.96), supporting the interpretation of convergent correlations. The shortened version of the TCU-ORC was found to have acceptable convergent validity with the ORCA scale ($r$=0.56, p<0.05), with a moderate correlation with the CARI ($r$=0.39, p<0.05). Convergent validities of $r$=0.70 are considered ideal, with validities less than $r$=0.50 considered insufficient in organisational research.[54 56]

## DISCUSSION
Validated measures for assessing organisational functioning and dynamics in health organisations in LMICs are currently lacking.[57] These measures are needed to assist health systems as they grapple with the need to be resilient and responsive to the changing health needs of the population.[6 32] To be applicable and useful, measures must be pragmatic and useful to the service providers involved in delivering and managing systems and facilities.[38 39] This study presents findings on the psychometric properties of the TCU-ORC among a sample of health workers from primary care settings in the Western Cape Province of South Africa. The sample was drawn from front-line providers in health facilities, who are key to influencing implementation of evidence-based practices through their role as 'street level bureaucrats'[58] who through their work have influence on the services received by facility clients. While the usefulness of this measure has been investigated by studies from a number of HICs,[59 60] this is the first investigation of its kind in primary care in South Africa and to our knowledge, in other LMICs.

The overall goal of this work was to develop a contextually relevant measure, feasible for use in primary care within South Africa.[35] While the original measure has been widely used in better resourced settings, 125 items is not a feasible for use as a routine assessment in the busy and under-resourced South African health system. A shorter, pragmatic instrument capable of collecting essential information to inform implementation is required for completion by primary care staff with high workloads and multiple competing priorities.[14] The need for reducing response burden and respondent fatigue is aligned with current thinking among global mental health researchers who have noted response burden as a threat to the quality of data collected by lengthy measures.[61] The current study resulted in the reduction of the TCU-ORC to 26 items. While this is a considerable reduction, further shortening may be needed to produce a scale truly feasible for routine use in primary care in other settings. Experience from HIC settings underscores the importance of ensuring implementation support measures are responsive to stakeholders' needs, present a low burden and have wide application across a multitude of settings.[38 39 62] Scales with large number of items have been shown to be feasibly shortened to as low as three items per subscale retaining high degree of validity and reliability (alpha >0.9).[63] With this in mind, we plan to conduct a Delphi study with stakeholders and experts from the South African primary care arena to further shorten the tool and enable integration of stakeholder

**Table 2** Principal axis factoring by domains of the original TCU-ORC

Subscale **A: Motivational readiness for change (29 items included in factor analysis)**
subscale **B: Institutional resources (22 items included in factor analysis)**
Subscale **C: Staff attributes (24 items included in factor analysis)**
subscale **D: Organisational climate (29 items included in factor analysis)**

| Initial factor 1 | | Final factor 1 Clinic organisational climate | | Initial factor 2 | | Final factor 2 Motivational readiness for change | | Initial factor 3 | | Final factor 3 Individual change efficacy | | Initial factor 4 | |
|---|---|---|---|---|---|---|---|---|---|---|---|---|---|
| Items | Loadings | Items | Loadings | Items | Loadings | Items | Loadings | Items | Loadings | Items | Loadings | Items | Loadings |
| A26 | 0.32 | B1 | 0.66 | A1 | 0.56 | A1 | 0.68 | B20 | 0.35 | C8 | 0.68 | B2 | 0.38 |
| B1 | 0.64 | B3 | 0.61 | A2 | 0.51 | A3 | 0.77 | C1 | 0.44 | C9 | 0.71 | C24 | 0.41 |
| B3 | 0.53 | B10 | 0.68 | A3 | 0.55 | A4 | 0.78 | C2 | 0.30 | C10 | 0.76 | D5 | 0.36 |
| B4 | 0.39 | B16 | 0.71 | A4 | 0.59 | A5 | 0.77 | C4 | 0.48 | C11 | 0.62 | D12 | 0.35 |
| B5 | 0.56 | B19 | 0.65 | A5 | 0.63 | A6 | 0.73 | C8 | 0.50 | C18 | 0.62 | D15 | 0.34 |
| B6 | 0.51 | C7 | 0.66 | A6 | 0.59 | A7 | 0.70 | C9 | 0.61 | | | D16 | 0.34 |
| B7 | 0.60 | D7 | 0.62 | A7 | 0.57 | A8 | 0.70 | C10 | 0.52 | | | D19 | 0.44 |
| B8 | 0.59 | D11 | 0.65 | A8 | 0.58 | A10 | 0.64 | C11 | 0.47 | | | D22 | 0.54 |
| B10 | 0.69 | D14 | 0.66 | A9 | 0.61 | | | C12 | 0.34 | | | D27 | 0.33 |
| B11 | 0.64 | D21 | 0.70 | A10 | 0.69 | | | C13 | 0.31 | | | | |
| B12 | 0.39 | D25 | 0.72 | A11 | 0.63 | | | C14 | 0.35 | | | | |
| B13 | 0.44 | D26 | 0.66 | A12 | 0.64 | | | C17 | 0.44 | | | | |
| B14 | 0.48 | D28 | 0.64 | A13 | 0.56 | | | C18 | 0.53 | | | | |
| B15 | 0.39 | D29 | 0.61 | A14 | 0.55 | | | C23 | 0.45 | | | | |
| B16 | 0.70 | | | A15 | 0.60 | | | D6 | 0.33 | | | | |
| B17 | 0.58 | | | A16 | 0.59 | | | | | | | | |
| B18 | 0.52 | | | A18 | 0.30 | | | | | | | | |
| B19 | 0.62 | | | A19 | 0.50 | | | | | | | | |
| B21 | 0.45 | | | A20 | 0.46 | | | | | | | | |
| C3 | 0.44 | | | A21 | 0.49 | | | | | | | | |
| C5 | 0.34 | | | A22 | 0.37 | | | | | | | | |
| C6 | 0.53 | | | A23 | 0.51 | | | | | | | | |
| C7 | 0.65 | | | A24 | 0.34 | | | | | | | | |
| C10 | 0.33 | | | A28 | 0.38 | | | | | | | | |
| C13 | 0.32 | | | A29 | 0.30 | | | | | | | | |
| C14 | 0.49 | | | C2 | 0.31 | | | | | | | | |

Continued

**Table 2** Continued

Subscale A: Motivational readiness for change (29 items included in factor analysis)
subscale B: Institutional resources (22 items included in factor analysis)
Subscale C: Staff attributes (24 items included in factor analysis)
subscale D: Organisational climate (29 items included in factor analysis)

| Initial factor 1 | | Final factor 1 Clinic organisational climate | | Initial factor 2 | | Final factor 2 Motivational readiness for change | | Initial factor 3 | | Final factor 3 Individual change efficacy | | Initial factor 4 | |
|---|---|---|---|---|---|---|---|---|---|---|---|---|---|
| Items | Loadings | Items | Loadings | Items | Loadings | Items | Loadings | Items | Loadings | Items | Loadings | Items | Loadings |
| C15 | 0.35 | | | C8 | 0.36 | | | | | | | | |
| C20 | 0.38 | | | C10 | 0.34 | | | | | | | | |
| C21 | 0.41 | | | C16 | 0.40 | | | | | | | | |
| C23 | 0.39 | | | C17 | 0.32 | | | | | | | | |
| D3 | 0.42 | | | C18 | 0.43 | | | | | | | | |
| D4 | 0.53 | | | D6 | 0.45 | | | | | | | | |
| D5 | 0.62 | | | D15 | 0.32 | | | | | | | | |
| D7 | 0.61 | | | D24 | 0.36 | | | | | | | | |
| D8 | 0.67 | | | | | | | | | | | | |
| D9 | 0.64 | | | | | | | | | | | | |
| D11 | 0.60 | | | | | | | | | | | | |
| D13 | 0.56 | | | | | | | | | | | | |
| D14 | 0.67 | | | | | | | | | | | | |
| D16 | 0.40 | | | | | | | | | | | | |
| D17 | 0.50 | | | | | | | | | | | | |
| D18 | 0.55 | | | | | | | | | | | | |
| D20 | 0.45 | | | | | | | | | | | | |
| D21 | 0.67 | | | | | | | | | | | | |
| D23 | 0.45 | | | | | | | | | | | | |
| D25 | 0.70 | | | | | | | | | | | | |
| D26 | 0.63 | | | | | | | | | | | | |
| D28 | 0.61 | | | | | | | | | | | | |

Extraction method: factor analysis.
Rotation method: oblique oblimin.
TCU-ORC, Texas Christian University Organisational Readiness for Change Scale.

**Table 3** Excluded items and proposed revised domains*

| Original TCU-ORC domain | No of items in initial factor analysis (104 total) | Excluded items (loadings <0.6 through 11 extractions) | Proposed revised domains | No of items in proposed scale |
|---|---|---|---|---|
| A Motivational Readiness for Change | 29 | Your facility needs guidance in<br>A2 using patient assessments to guide clinical care<br>A9 defining its mission<br>A11 assigning or clarifying staff roles<br>A12 setting out accurate job descriptions for staff<br>A13 evaluating staff performance<br>A14 improving relations among staff<br>A15 improving communications among staff<br>A16 improving record keeping and information systems<br>A17 basic computer skills/programmes<br>A18 specialised computer applications<br>A19 new developments and practices in your area of responsibility<br>A20 new procedures being used or planned<br>A21 continuous professional development<br>A22 new laws or regulations you need to know about<br>A23 management or supervisory responsibilities<br>Current pressures to make facility changes come from<br>A24 community members (patients)<br>A25 other staff members<br>A26 facility supervisors or managers<br>A27 district level managers<br>A28 community groups<br>A29 national and provincial DOH structures | **Motivational Readiness for Change**<br>Clinical staff at your facility needs guidance in:<br>A1 assessing patient needs<br>A3 using patient assessments to document patient improvements<br>A4 matching patient needs with services<br>A5 improving relationship with patients<br>A6 improving patient thinking and problem-solving skills to manage chronic disease<br>A7 improving behavioural management of patients<br>A8 identifying and using evidence-based practices<br>Your facility needs guidance in:<br>A10 setting specific goals for improving services | 8 |
| B Institutional Resources | 22 | B2 Frequent staff turnover is a problem for your facility<br>B4 The space in your facilities are adequate for conducting group or individual counselling<br>B5 You have clinical supervisors who are capable and qualified<br>B6 You learnt new skills or techniques at a professional training in the past year<br>B7 Much time and attention are given to staff supervision when needed<br>B8 Staff in your facility are able to spend the time needed with patients<br>B9 Equipment at your facility is mostly old and outdated<br>B11 Support staff in your facility have the skills they need to do their jobs<br>B12 Offices in your facility allow the privacy needed for individual counselling<br>B13 Your facility holds regular in-service training<br>B14 Your facility has enough staff to meet current patient needs<br>B15 Clinical staff in your programme are well-trained<br>B17 Offices and equipment in your facility are adequate<br>B18 Your facility provides a comfortable reception/waiting area for patients<br>B20 A larger support staff is needed to help meet needs at your programme.<br>B21 Staff in your facility are able to attend professional training<br>B22 Staff concerns are ignored in most decisions made in your facility<br>D29 You feel encouraged to try new and different techniques | **Clinic organisational climate**<br>B1 You have good facility management at your facility<br>B3 Staff training and continuing education are priorities in your facility<br>B10 Clinical and management decisions for your facility are well planned<br>B16 You have confidence in how decisions at your facility are made<br>B19 Staff meet frequently with clinical supervisors about patient needs and progress<br>C7 Your facility encourages and supports professional growth<br>D7 Ideas and suggestions in your facility get fair consideration by management<br>D11 Your facility staff is always kept well informed<br>D14 Your facility operates with clear goals and objectives<br>D21 The formal and informal communication channels in your facility work very well<br>D25 Management fully trusts professional judgments of staff in your facility<br>D26 Staff members always feel free to ask questions and express concerns in your facility<br>D28 Management for your facility has a clear plan for its future | 13 |

**Table 3** Continued

| Original TCU-ORC domain | No of items in initial factor analysis (104 total) | Excluded items (loadings <0.6 through 11 extractions) | Proposed revised domains | No of items in proposed scale |
|---|---|---|---|---|
| C Staff Attributes | 24 | C1 You have the skills needed to effectively perform your role C2 Other staff often ask your advice C3 You are satisfied with your present job C4 Learning and using new procedures are easy for you C5 You are considered an experienced source of advice about services and/ or facility management C6 You feel appreciated for the job you do at work C12 You regularly influence the decisions of other staff you work with C13 You usually accomplish whatever you set your mind on C14 You do a good job of regularly updating and improving your skills C15 You regularly read professional articles or books related to your work C16 Other staff often ask for your opinions about issues in the facility C17 You are willing to try new ideas even if some staff members are reluctant C19 You are sometimes too cautious or slow to make changes C20 You are proud to tell others where you work C21 You like the people you work with C22 You are viewed as a leader by the staff you work with C23 You consistently plan ahead and carry out your plans C24 You would like to find a job somewhere else | **Individual change efficacy** C8 You are effective and confident in doing your job C9 You are able to adapt quickly when you have to make changes C10 Keeping your professional skills up-to-date is a priority for you C11 You give high value to the work you do C18 You frequently share your knowledge of new ideas with others | 5 |
| D Organisational Climate | 29 | D1 Some staff members seem confused about the main goals for your facility D2 The heavy staff workload reduces the effectiveness of your facility D3 You frequently hear good ideas from other staff for improving services D4 Care plans and decisions for patients in your facility often get revised by a clinical supervisor D5 The general attitude in your facility is to accept new and changing technology or procedures D6 More open discussions about facility issues are needed where you work D8 Staff members at your facility work together as a team D9 Your duties are clearly related to the goals for your facility D10 You are under too many pressures to do your job effectively D12 New ways of working from staff are discouraged where you work D13 Mutual trust and cooperation among staff in your programme are strong D15 Staff members at your facility often show signs of high stress and strain D16 It is easy to change procedures at your facility to meet new conditions D17 Staff in your facility can try out different techniques to improve their effectiveness D18 Staff members at your facility get along very well D19 Staff members are given too many rules in your facility D20 Staff members at your programme are quick to help one another when needed D22 There is too much friction among staff members you work with D23 Staff members at your facility understand the work of the facility fits as part of the health improvement in your community D24 Some staff in your facility do not do their fair share of work D27 Staff frustration is common where you work | | |

*'A' items refer to the Motivational Readiness for Change scale, 'B' items refer to the Institutional Resources scale, 'C' items refer to the Staff Attributes scale, 'D' items refer to the Organisational Climate scale. DOH, Department of Health; TCU-ORC, Texas Christian University Organisational Readiness for Change Scale.

**Table 4** Internal consistency and retest reliability of the new subscales

| Subscale/factor | Description | Cronbach's alpha | Temporal stability (R) |
|---|---|---|---|
| Clinic organisational climate | Management practices, trust, leadership, communication, appreciation of staff, facility resources, attitudes to change | 0.90 | 0.87* |
| Motivational readiness for change | Areas of need for facility improvement and staff development | 0.91 | 0.60* |
| Individual change efficacy | Individual self-efficacy, adaptability, knowledge and skills | 0.83 | 0.53* |
| Total revised scale | | 0.86 | 0.80* |

*Significant at the 0.01 level.

perspectives. Subsequently a confirmatory factor analysis (CFA) will be conducted.

Factor analysis showed overlap between three of the original domains, suggesting a three-factor structure for the South African data, in contrast to the original four domains ((Motivational Readiness for Change (A), Institutional Resources (B) Staff Attributes (C) and Organisational Climate (D)). The proposed domains for the South African data, as shown in table 2 are (1) Clinic Organisational Climate, (13 items), (2) Pressures for Change (8 items) and (3) Individual change efficacy (5 items). Each domain and the overall scale showed strong internal consistency, adequate temporal stability, and moderate convergent validity with the ORCA and CARI measures. The temporal stability for the Individual Change Efficacy scale may reflect the unstable and uncertain climate within which health providers often operate. At the time of this study, many of the clinics were affected by community violence and service delivery protests that sometimes led to clinic closures or restricted service delivery. It is plausible that these contextual factors may have impacted on individual providers perceptions of change efficacy, especially as efficacy for change is often context dependent. However, more research is needed to identify and address such 'bridging factors' that demonstrate bidirectional influences of outer system and inner organisational contexts.[64]

Three items with cross loadings >0.3 (0.3–0.5) were removed suggesting retained items were representing relatively unique constructs that were understood by respondents.[57] The domain 'Clinic Organisational Climate' incorporates items from the Institutional Resources, Staff Attributes and Organisational Climates domains of the original TCU-ORC. This may be because institutional resources, staff attributes, and trusting supervisory relationships become essential prerequisites for staff's functioning within constrained environments. These are perceived as key aspects of the organisational climate and have been previously shown to group as elements of organisational functioning.[10] Thirteen items were retained in this domain suggesting these are key to assessing organisational climate in this setting. Items included those relating to management, supervision, teamwork, leadership and trusting relationships, which

aligns with other work investigating primary care functioning in South Africa.[19 32 65 66] Findings from qualitative work assessing primary care facilities' preparedness for mental health counselling implementation also suggest intersecting relationships between resource availability, management style and facility environment.[21] The Pressures for Change domain remained largely consistent with the corresponding domain of the original TCU-ORC. Only a small number of items were retained from the C domain on Staff Attributes—these related to individual agency for change. Organisational theory outlines how individuals within an organisation can be resistant to change, particularly in the context of heavy workloads[27 67] known to be present in South African facilities. South African facility managers are often clinicians, without management training, who work in high workload environments and indicate that workload constrains innovation adoption.[68] The South African health system has a strong hierarchy of downward supervision and management, compliance and targets and this may hinder frontline managers and staff from being responsive to patient needs and public health burden.[69] At the same time, nurses are the majority of clinic staff and are key to organisational climate and change. The items retained in this scale may, therefore, be most relevant for assessing individual perceptions among nursing staff on their agency and willingness to change.

In relation to convergent validity, correlation of TCU-ORC-SF with the ORCA showed large effect, but with the CARI correlation showed medium effect. These two additional instruments were also designed to assess ORC in HICs, and although some aspects of organisational culture, dynamics and functioning are translatable to other settings, others may not be robustly relevant in the South African PHC context. With regard to correlation of the shortened tool with the CARI, it is likely that the restricted range of responses in the CARI resulted in a lower correlation.[70] The CARI items are measured on a 4-point scale, while the TCU-ORC items are measured on a 5-point scale. Nonetheless, it is the opinion of the researcher/s that these validity correlations between the ORCA, CARI and TCU-ORC scales are acceptably large to assume convergence. We propose that the revised domains presented here more closely capture the essentials of

ORC in South African PHC settings than the original measures. This adapted measure may enable progress on assessing readiness for implementation of evidence-based practices and addressing barriers in South African health facilities, identified as a prerequisite for implementation in our previous work.[10] Further, there will be a wide variety of factors influencing readiness for change, particularly in different settings in LMIC and elsewhere, and detailed implementation research elucidating these elements would support the use or adaptation of this tool in different settings. The current work provides a base for driving policy implementation to enable adoption of innovations to improve public health through strengthening PHC platforms.

## Limitations

Data for this study were collected from primary care facilities from the Western Cape province. This province is more well resourced than other provinces. Data from the study showed a sociodemographic profile of facility staff that reflects the qualifications, professions and work experience common across primary care settings in the country. This suggests appropriateness for the broader South African context, although there are important disparities in human and other resources and variations in burden of disease and contexts across provinces[5 71] and additional adaptations could be required in other areas. The survey was in English which is not the first language of health providers in some settings, however, English is the official business language of government agencies and health providers would be expected to be proficient. The ORCA and CARI measures have also not been adapted for use in LMIC use so there may be limitations in their validity for this context which may have influenced convergent validity findings.

## CONCLUSION

This study has presented the contextual adaptation of the TCU-ORC in an LMIC setting. Based on the findings from this study, we propose a short form of the TCU-ORC, comprising 26 of 125 items. The three subscales proposed: Clinic Organisational Climate, Motivational Readiness for Change, and Individual Change Efficacy contain key items for assessing ORC in this context. This scale may be useful as a tool for health service managers to highlight potential roadblocks to the introduction of innovations that may need to be addressed when developing an implementation strategy. This shortened tool, however, may still be too lengthy for widespread practical use in primary care services and further shortening involving stakeholders who will use the tool in practice is required.

### Author affiliations
[1]Alcohol Tobacco and Other Drug Research Unit, South African Medical Research Council, Tygerberg, South Africa
[2]Alan J Flisher Centre for Public Mental Health, Department of Psychiatry and Mental Health, University of Cape Town, Cape Town, South Africa
[3]Division of Addiction Psychiatry, Department of Psychiatry and Mental Health, University of Cape Town, Cape Town, South Africa
[4]Faculty of Community and Health Sciences, University of the Western Cape, Cape Town, South Africa
[5]Department of Psychiatry, University of California, La Jolla, San Diego, California, USA
[6]Child and Adolescent Services Research Center, San Diego, California, USA

**Acknowledgements** The authors thank Lesley-Anne Erasmus-Claasen, Yuche Jacobs and Nombuso Moshiga for their fieldwork on the study as well as all participating facilities, managers and staff.

**Contributors** CB-S and BM conceptualised the study. CB-S and PP-W managed data collection. BM and KS are principal investigators on the Project MIND trial in which the study is nested. CB-S and EW conducted factor analysis. EW provided statistical technical input and drafted some sections of the results. GAA provided technical guidance on methods and analysis. All authors reviewed and contributed to the final version of the manuscript. CBS is responsible for the overall content as the guarantor.

**Funding** This study is funded jointly by the British Medical Research Council, Wellcome Trust, Department for International Development, the Economic and Social Research Council, the Global Challenges Research Fund (MR/M014290/1) as well as funding from the South African Medical Research Council (SAMRC, contract). CB-S was supported as a postdoctoral fellow by funding from the SAMRC through its Division of Research Capacity Development under the Intra-Mural Postdoctoral Fellowship Programme from funding received from the South African National Treasury.

**Disclaimer** The content hereof is the sole responsibility of the authors and does not necessarily represent the official views of the SAMRC or the funders.

**Competing interests** None declared.

**Patient consent for publication** Not applicable.

**Ethics approval** Ethical approval was granted for this study by the Human Research Ethics Committee of the South African Medical Research Council (EC004-2-2015) and approval for working in facilities was also granted by the Western Cape Provincial Department of Health.

**Provenance and peer review** Not commissioned; externally peer reviewed.

**Data availability statement** Data are available on reasonable request. Extra data can be accessed via the Dryad data repository at http://datadryad.org/ with the doi.org/10.5061/dryad.w6m905qqm.

**ORCID iD**
Carrie Brooke-Sumner http://orcid.org/0000-0002-9489-8717

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
