## [Reviewer comments · BMJ Open]

ARTICLE DETAILS

TITLE (PROVISIONAL)	Adaptation of the Texas Christian University Organisational Readiness for Change Short Form (TCU-ORC-SF) for use in primary health facilities in South Africa
AUTHORS	Brooke-Sumner, Carrie; Petersen-Williams, Petal; Wagener, Emma; Sorsdahl, Katherine; Aarons, Gregory; Myers, Bronwyn

VERSION 1 – REVIEW

REVIEWER	Pretestini, Anna University of Milan, Department of Economy, Management and Quantitative Methods
REVIEW RETURNED	19-Dec-2020

GENERAL COMMENTS	The paper is focused on the validation of a shortened tool for use in primary health facilities in South Africa. From a methodological point of view, it seems clear and complete. In the background section the authors discussed the importance to apply instruments such as TCU-ORC in order to discover what kind of factors can support or hinder initial adoption of new evidence-based practices. So, thanks the huge amount of data and information that they have collected for the validation, I would suggest adding something more about the specific results from the 24 facilities involved. Otherwise, the authors can explain if they want to use the already collected data for other scopes than the validation of the study. In fact, the topic of the paper can be really interesting for the implications on future research but also for its impact on current managerial practices.
--

REVIEWER	Okonofua , F University of Benin, Obstetrics and Gynaecology
REVIEW RETURNED	25-Jan-2021

GENERAL COMMENTS	Originality The paper reports the results of a study that assessed factors influencing the adoption of the TCU-ORC-SF in South Africa. The change was developed in Texas and had been used in high-income countries, without evidence that it would be useful in LMICs. This study was specifically designed to test its applicability in South Africa and identify the validity of a shorter version. It is critically important to assess the readiness for change especially when new procedures or clinical guidelines are being introduced. However, the change mantra is culturally specific – what would work in one context may not necessarily work in other settings. Thus, the results of this study would only apply to primary health care in South Africa and not necessarily to other LMICs. The fact that the study was carried out in
--

	a more prosperous part of South Africa further helps to authenticate this belief and raises the issue of external validity of the results even within South Africa itself. Furthermore, because of the multiplicity of factors that influence readiness for change, it is likely that only a properly conducted implementation research that examines all contextual elements may be effective in identifying the change readiness. It would be relevant for the authors to discuss these aspects in the paper. Scientific Merit I have additional comments as follows:  • In the abstract, the first five lines should be presented in the Methods section. Indeed, there is currently no methods section in the abstract. • The Project MIND was first used in the introduction, and then Project MINF was mentioned later. The authors have to decide the correct version and also provide its full meaning when first used. • It would be relevant to describe how the 24 primary health facilities were selected. Was this random? It will be relevant to provide some information about primary health care in South Africa, how many are available? so the appropriateness of the sample used can be understood. The organization of primary health care services in the country also needs to be described, especially to provide an understanding on relevant stakeholders involved in the management of change. • My major concern with the study is the fact that these are self-reporting of intentions by midwives. It does not necessarily mean that they would implement what they have reported in real life situation. This aspect needs to be discussed as a limitation of the study. It is also not clear whether the nurses interviewed are the main decisionmakers in the primary health care system in South Africa. The non-involvement of decisionmakers raises questions as to the usefulness of the model in real life. This aspect also needs to be discussed in the paper. • The discussion part of the paper should be properly delineated. • Also, the policy implications and public health relevance of the study should be better presented. • There are several typos and grammatical errors in various parts of the paper that need to be corrected.
--	--

REVIEWER	Gregory, Megan The Ohio State University
REVIEW RETURNED	06-May-2021

GENERAL COMMENTS	I appreciated the opportunity to statistically review the manuscript "Validation of the Texas Christian University Organisational Readiness for Change Short Form (TCU-ORC-SF) for use in primary health facilities in South Africa." The study seeks to validate a short form of an organizational readiness self-report measure in low- and middle-income countries. The authors re-administered the test after 2 weeks to assess test-retest reliability, computed Cronbach alphas, conducted an EFA, and examined convergent validity by assessing correlations with similar measures. For reference, I also read the study protocol paper cited by the authors in the Method section(1) to obtain sufficient information for this review. The authors described their methods and decisions in depth, and I appreciated the detail in Supplementary table 1 explaining the rationale for removing some of the long form items. The EFA methods used (principal axis factoring, oblique oblimin rotation) are
---

appropriate for these data. The scale shows adequate test-retest reliability, internal consistency reliability, and convergent validity. However, I had some concerns, as indicated below:

Major Concerns and Questions:

1. A subsequent confirmatory factor analysis (CFA) should be performed to confirm the factor structure that emerged in the EFA(2). In fact, this is what the authors proposed to do in their study protocol paper. The CFA should be done with a separate sample. Given the logistical challenges with this sample, the authors could first explore whether they could randomly split the current sample and run an EFA on one half and a CFA on the other, to determine if there is adequate statistical power to do so, without an unstable and potentially inaccurate model. This is of more concern due to the way the authors came about the number of factors in the EFA- going from 6 which is what the parallel analysis suggested, to deciding to try 4 based on theory, and ending up with 3. CFA would help inform which of these models was the best fit. At an absolute minimum, lack of CFA should be mentioned as a limitation and this should be stated as a next step in future research. However, I would still have some concerns here, although I do understand this sample is unique and it could be challenging to gain a much larger sample size.
2. The original scale contained four domains, whereas the short form EFA found only three. I would request that the authors create a table to map the short form items and domains onto the original scale's items and domains, to allow for assessment of whether there is construct deficiency in the short form. I have additional concerns about this given that the authors retained only those items that loaded above .60, instead of the conventional .30-.40. Was an important aspect of the construct (e.g., potentially those items that represent one of the subdomains on the original scale) possibly lost due to this decision? A supplementary table that lists every item (providing the full wording of the item and its alphanumeric code used in the manuscript, e.g., A26), whether the wording was adapted and how, its domain on the original scale, whether it was excluded by the Delphi process, its factor loadings (before removing items <.60), and its domain on the final scale (if any), would be helpful. Relatedly, P. 10, lines 203-204: Can you clarify what is meant by "factor extraction was determined on the basis of the theoretical structure of the TCU-ORC" and p. 11 lines 240-244 "the decision was made to specify four factors"? The way it is currently written, it does not seem particularly data-driven, in the spirit of EFA. It would help to see the output. P. 11, line 235 – is 74 factors a typo? That seems enormous. Would the authors be willing to share the output from this EFA? P. 11 line 236 – should be spelled Glorfeld, not Glorefeld. P. 11 line 236 - I would suggest re-running the PA with Glorfeld's extension at the 95th percentile to see if results differ.
3. P. 9 indicates that cases with >50% missing data were excluded. What was done with missing data for the remaining cases, particularly for the EFA?

More Minor Issues:

4. There is a typo on p. 8, line 162 – alpha should be listed as > .50, not >.05, confirmed by reviewing Table 1 of the cited reference(3). Although alphas below >.70 on the original scale are not optimal, I understand that most of the original scale has alphas above this threshold, with just a few subdomains appearing to be below this. In addition, the authors of the current study found alphas well above .70 for their modified scale, so I believe this is not a major issue.
5. The table note for Table 2 seems to have an error. Extraction

	method should be Principal axis factoring, not factor analysis. 6. Because this paper simultaneously created a short form and adapted the measure for cultural modification specific to the nuances of South Africa, future work should be done to determine whether validity of this new short form holds in other countries, or if other cultural modifications would need to be done. For example, if researchers in the U.K., wanted to utilize this short form, it might be the case that validity may not hold due to initial removal of the items about using computers and internet at work for the current study. This is acknowledged a bit, I think, by the title and paper stating that this short form is for “low and middle income countries,” however, this should also be mentioned in the limitations. Along these lines, the paper seems to sway back and forth regarding if this is generalizable to only South Africa, or to all low and middle income countries. Is there reason to believe it is, in fact, generalizable outside of South Africa? 7. There should be a table showing the final retained items, by their new domains. Table 3 (while informative in its own right) doesn’t really achieve this, as it maps items to the original four factors, rather than the three final factors. References  1. Brooke-Sumner C, Sorsdahl K, Lombard C, Petersen-Williams P, Myers B. Protocol for development and validation of a context-appropriate tool for assessing organisational readiness for change in primary health clinics in South Africa. BMJ Open. 2018;8(4): e020539. 2. Hinkin TR. A brief tutorial on the development of measures for use in survey questionnaires. Organizational Research Methods. 1998;1(1):104-21. 3. Lehman WE, Greener JM, Simpson DD. Assessing organizational readiness for change. Journal of Substance Abuse Treatment. 2002;22(4):197-209.
--	---

VERSION 1 – AUTHOR RESPONSE

Reviewer 1 Comment	Response
In the background section the authors discussed the importance to apply instruments such as TCU-ORC in order to discover what kind of factors can support or hinder initial adoption of new evidence-based practices. So, thanks the huge amount of data and information that they have collected for the validation, I would suggest adding something more about the specific results from the 24 facilities involved. Otherwise, the authors can explain if they want to use the already collected data for other scopes than the validation of the study. In fact, the topic of the paper can be really interesting for the implications on future research but also for its impact on	Thank you for this comment, we have now included some information from our previous work, in the background section (p 4) which speaks to the adoption of this new practice in our context. We have also referred to these earlier findings in the discussion section (p 18).

current managerial practices.	
Reviewer 2 Comment	Response
It is critically important to assess the readiness for change especially when new procedures or clinical guidelines are being introduced. However, the change mantra is culturally specific – what would work in one context may not necessarily work in other settings. Thus, the results of this study would only apply to primary health care in South Africa and not necessarily to other LMICs. The fact that the study was carried out in a more prosperous part of South Africa further helps to authenticate this belief and raises the issue of external validity of the results even within South Africa itself. Furthermore, because of the multiplicity of factors that influence readiness for change, it is likely that only a properly conducted implementation research that examines all contextual elements may be effective in identifying the change readiness. It would be relevant for the authors to discuss these aspects in the paper.	Thank you for these comments, we have included this in the limitations section (p 18) and in the discussion (p 18).
In the abstract, the first five lines should be presented in the Methods section. Indeed, there is currently no methods section in the abstract.	This has been incorporated (p 2).
The Project MIND was first used in the introduction, and then Project MINF was mentioned later. The authors have to decide the correct version and also provide its full meaning when first used.	Thank you for this, it was a typo and has been addressed (p 6, 7)
It would be relevant to describe how the 24 primary health facilities were selected. Was this random? It will be relevant to provide some information about primary health care in South Africa, how many are available? so the appropriateness of the sample used can be understood. The organization of primary health care services in the country also needs to be described, especially to provide an understanding on relevant stakeholders involved in the management of change.	Thank you for this comment. Information on this selection (in relation to the trial in which the study was nested) has now been provided.
My major concern with the study is the fact that these are self-reporting of intentions by midwives. It does not necessarily mean that they would implement what they have reported in real life situation. This aspect needs to be discussed as a limitation of the study. It is also not clear whether	Thank you for this point on decision making and leadership. We agree that nurses are not the level of decision makers who control resources and policy directives. However, there is work in South Africa that indicates their influence on implementation of policies through their action or

the nurses interviewed are the main decisionmakers in the primary health care system in South Africa. The non-involvement of decisionmakers raises questions as to the usefulness of the model in real life. This aspect also needs to be discussed in the paper.	inaction as 'Street level bureaucrats' who are those working on the 'frontlines' and who influence implementation through their work. We have included reference to this work (p 15) and how this indicates the appropriateness of the sample.
The discussion part of the paper should be properly delineated.	Discussion section is indicated on page 14.
Also, the policy implications and public health relevance of the study should be better presented.	Thank you for this comment, we have added on p18.
There are several typos and grammatical errors in various parts of the paper that need to be corrected.	Noted, thank you.
Reviewer 3	Response
A subsequent confirmatory factor analysis (CFA) should be performed to confirm the factor structured that emerged in the EFA(2). In fact, this is what the authors proposed to do in their study protocol paper. The CFA should be done with a separate sample. Given the logistical challenges with this sample, the authors could first explore whether they could randomly split the current sample and run an EFA on one half and a CFA on the other, to determine if there is adequate statistical power to do so, without an unstable and potentially inaccurate model. This is of more concern due to the way the authors came about the number of factors in the EFA- going from 6 which is what the parallel analysis suggested, to deciding to try 4 based on theory, and ending up with 3. CFA would help inform which of these models was the best fit. At an absolute minimum, lack of CFA should be mentioned as a limitation and this should be stated as a next step in future research. However, I would still have some concerns here, although I do understand this sample is unique and it could be challenging to gain a much larger sample size.	Thank you for this thoughtful and thorough review and referral to the study protocol. We have noted these concerns and respond as follows:  • There will be several phases to the study – this initial stage should be referred to as an adaptation and reduction of items for feasibility, not a validation, as the reviewer has pointed out. We have changed this throughout. • The next step will be the Delphi approach as indicated • We have now clarified that the CFA will be conducted as the subsequent stage of the research • We have also clarified that the goal of this adaptation was to reduce the questions to a smaller, more manageable and relevant questionnaire (p 6).
The original scale contained four domains, whereas the short form EFA found only three. I would request that the authors create a table to map the short form items and domains onto the original scale's items and domains, to allow for assessment of whether there is construct deficiency in the short form.	Thank you for this suggestion and the concern re construct deficiency. We refer to the point above where we have reframed the wording to our original intention with the study, of adaptation rather than validation. With this in mind we feel that Table 3 with excluded items and proposed revised domains (including wording) should be

I have additional concerns about this given that the authors retained only those items that loaded above .60, instead of the conventional .30-.40. Was an important aspect of the construct (e.g., potentially those items that represent one of the subdomains on the original scale) possibly lost due to this decision? A supplementary table that lists every item (providing the full wording of the item and its alphanumeric code used in the manuscript, e.g., A26), whether the wording was adapted and how, its domain on the original scale, whether it was excluded by the Delphi process, its factor loadings (before removing items <.60), and its domain on the final scale (if any), would be helpful. Relatedly, P. 10, lines 203-204: Can you clarify what is meant by “factor extraction was determined on the basis of the theoretical structure of the TCU-ORC” and p. 11 lines 240-244 “the decision was made to specify four factors”? The way it is currently written, it does not seem particularly data-driven, in the spirit of EFA. It would help to see the output. P. 11, line 235 – is 74 factors a typo? That seems enormous. Would the authors be willing to share the output from this EFA? P. 11 line 236 – should be spelled Glorfeld, not Glorefeld. P. 11 line 236 - I would suggest re-running the PA with Glorfeld’s extension at the 95th percentile to see if results differ.	sufficient. Regarding the stringent condition of loading to .60, which fits with the study goal of adaptation and data reduction as opposed to validation. Please note that the Delphi results are not presented in this paper, and the supplementary table we have included does show the items that were excluded based on lack of relevance for the study context (based on research team discussion). The wording of the remaining items is contained in Table 3. Noted, we have included the output as an attachment.
---	--

Thank you for the spelling correction.

We include below text that was in our original manuscript but that was removed for length consideration. Given the potential readership of BMJOpen we decided this level of detail could be removed for space considerations.

Factor extraction. The initial extraction was run with the Kaiser rule specified, retaining factors with eigenvalues greater than 0. This resulted in 74 factors identified. A parallel analysis was run in STATA, using Horn's parallel analysis as a post-estimation command, with 1000 repetitions specified. This result suggested that 17 factors be retained, as indicated by the number of factors observed in which eigenvalues were greater than the randomly generated eigenvalues. This was still considered to be too many factors, based on the theoretical structure of the measure, as well as the goal to reduce the number of items. It has been noted that this method of estimation has the tendency to over-extract factors, as it uses the 50th percentile (or the mean) to compare eigenvalues. It was decided to run the parallel analysis again, and separately, using Glorefeld's (1995) extension and suggestion of using the 99th percentile to increase sensitivity. The command was run using the method employed by Dinno (2009), revealing that 15 factors had adjusted eigenvalues greater than those randomly generated .

It has been established that factors with a large number of variables are prone to inaccuracies due to poorly defined factors (Jones, 2018). Due to the large number of variables in this measure, it is likely that the results of the PA are over extracting factors. The decision was ultimately made to specify four factors, according to these results and theoretical basis of the four subscales of the original TCU-ORC (40). After the initial extraction of four factors, we observed that multiple items from two subscales loaded onto one factor. Visual inspection of the scree plot confirmed three distinct points above the point of inflection (49), suggesting a three-factor solution. Subsequent analyses were based on the

3. P. 9 indicates that cases with >50% missing data were excluded. What was done with missing data for the remaining cases, particularly for the EFA?	specification of a three-factor solution. STATA will have excluded listwise.
There is a typo on p. 8, line 162 – alpha should be listed as > .50, not >.05, confirmed by reviewing Table 1 of the cited reference(3). Although alphas below >.70 on the original scale are not optimal, I understand that most of the original scale has alphas above this threshold, with just a few subdomains appearing to be below this. In addition, the authors of the current study found alphas well above .70 for their modified scale, so I believe this is not a major issue.	Thank you for this, we have corrected.
The table note for Table 2 seems to have an error. Extraction method should be Principal axis factoring, not factor analysis.	Thank you, corrected.
Because this paper simultaneously created a short form and adapted the measure for cultural modification specific to the nuances of South	Thank you for this comment, which was also brought up by another reviewer. We have made our position regarding wider relevance of the

Africa, future work should be done to determine whether validity of this new short form holds in other countries, or if other cultural modifications would need to be done. For example, if researchers in the U.K., wanted to utilize this short form, it might be the case that validity may not hold due to initial removal of the items about using computers and internet at work for the current study. This is acknowledged a bit, I think, by the title and paper stating that this short form is for “low and middle income countries,” however, this should also be mentioned in the limitations. Along these lines, the paper seems to sway back and forth regarding if this is generalizable to only South Africa, or to all low and middle income countries. Is there reason to believe it is, in fact, generalizable outside of South Africa?	measure clearer now (p 18). We hope this is addresses this comment.
There should be a table showing the final retained items, by their new domains. Table 3 (while informative in its own right) doesn’t really achieve this, as it maps items to the original four factors, rather than the three final factors	Thank you for this suggestion. Again, we made Table 3 as it is to assist in keeping the paper concise, as the relation to the original factors is covered in the discussion.

VERSION 2 – REVIEW

REVIEWER	Gregory, Megan The Ohio State University
REVIEW RETURNED	18-Aug-2021

GENERAL COMMENTS	I appreciated the authors reframing of the paper from validation to adaptation, which seems more accurate given lack of follow-up CFA. In addition, they have now mentioned that a CFA will be a next step, and that the scale may require other adaptations for other settings. However, some concerns remain from my original review, as noted below. I am not sure if there is a reviewer response letter that addresses the below; if so, I did not have access to it:  1. P.10 indicates that cases with >50% missing data were excluded. What was done with missing data for the remaining cases? 2. P. 11 - I would suggest re-running the PA with Glorfeld’s extension at the 95th percentile to see if results differ.
--

VERSION 2 – AUTHOR RESPONSE

Response letter re: bmjopen-2020-047320.R2

Adaptation of the Texas Christian University Organisational Readiness for Change Short Form (TCU-ORC-SF) for use in primary health facilities in South Africa.

Thank you for allowing us to opportunity to submit this revision. Please see below the author responses to the comments provide by reviewer 3.

Reviewer 3 comments	Author Response
1. P.10 indicates that cases with >50% missing data were excluded. What was done with missing data for the remaining cases?	Cases with missing values would be excluded listwise from all subsequent analysis (this is the standard setting in STATA). This has been added to manuscript (pg. 9. line 201 in 'clean' copy).
2. P. 11 - I would suggest re-running the PA with Gorfeld's extension at the 95th percentile to see if results differ.	We have included the parallel analysis output as an attachment. The outputs for parallel analyses using the mean, 95th and 99th percentile have been added to this document. The resulting factors that the various PA's suggested were all higher than theoretically relevant and manageable. The 99th percentile extension suggested 15 factors, and the 95th percentile extension suggested 14 factors to retain. It was decided to proceed with the theoretical 4 factors of the measure. The addition of using the 95th percentile was added to the manuscript. (pg. 11, lines 238-241 in 'clean' copy)